# Long-Term Examination of Degradation and In Vivo Biocompatibility of Some Mg-0.5Ca-xY Alloys in Sprague Dawley Rats

**DOI:** 10.3390/ma15175958

**Published:** 2022-08-29

**Authors:** Ștefan Lupescu, Corneliu Munteanu, Eusebiu Viorel Sindilar, Bogdan Istrate, Iuliana Mihai, Bogdan Oprisan, Aurelian-Sorin Pasca

**Affiliations:** 1Department of Mechanics and Technologies, Stefan cel Mare University of Suceava, 13 University Street, 720229 Suceava, Romania; 2Mechanical Engineering Department, Gheorghe Asachi University of Iasi, 6 D. Mangeron Blvd, 700050 Iasi, Romania; 3Technical Sciences Academy of Romania, 26 Dacia Blvd, 030167 Bucharest, Romania; 4Faculty of Veterinary Medicine of Lasi, “Ion Ionescu de la Brad” Iași University of Life Sciences (IULS), nr.8, Mihail Sadoveanu Alley, 700490 Iasi, Romania; 5Faculty of Medicine, “Grigore T. Popa” University of Medicine and Pharmacy from Iasi, Universității 16 Street, 700115 Iasi, Romania

**Keywords:** Mg-Ca-Y biodegradable alloys, surface morphology, in vivo animal study

## Abstract

The medical field has undergone constant development in recent years, and a segment of this development is occupied by biodegradable alloys. The most common alloys in this field are those based on Mg, their main advantage being the ability to degrade gradually, without affecting the patient, and also their ability to be fully absorbed by the human body. One of their most important conditions is the regeneration and replacement of human tissue. Tissue can be engineered in different ways, one being tissue regeneration in vivo, which can serve as a template. In vivo remodeling aims to restore tissue or organs. The key processes of tissue formation and maturation are: proliferation (sorting and differentiation of cells), proliferation and organization of the extracellular matrix, biodegradation of the scaffold-remodeling, and potential tissue growth. In the present paper, the design of the alloys in the Mg-Ca-Y system is formed from the beginning using high-purity components, Mg-98.5%, master-alloys: Mg-Y (70 wt.%–30 wt.%) and Mg-Ca (85 wt.%–15 wt.%). After 8 weeks of implantation, the degradation of the implanted material is observed, and only small remaining fragments are found. At the site of implantation, no inflammatory reaction is observed, but it is observed that the process of integration and reabsorption, over time, accentuates the prosaic surface of the material. The aim of the work is to test the biocompatibility of magnesium-based alloys on laboratory rats in order to use these alloys in medical applications. The innovative parts of these analyses are the chemical composition of the alloys used and the tests performed on laboratory animals.

## 1. Introduction

Biodegradable alloys are increasingly being developed and used in the medical field [1]. Rare earth elements having a significant positive influence in this industry are of major interest in specialized studies. The main advantages of these biodegradable alloys are their ability to gradually degrade over time without affecting the patient’s condition and, most importantly, their capacity to be fully absorbed by the human body. The most widespread and widely used biodegradable alloys are based on magnesium, an essential element found in the human body [2].

In the medical field, these biodegradable alloys are used in the form of heart valves, sutures, fracture plates, and orthodontic wires. In recent years, both the constant aging of the population and the lack of an active lifestyle have led to an increase in bone-related diseases and bone fractures [3].

However, in order to be accepted by the human body, these biodegradable alloys must meet certain conditions. One of the most important ones, which plays a very important role, is the replacement and regeneration of human tissue. In short, the materials should provide a matrix to support cell attachment and migration, in order to generate new bone in a process called osteoconduction, and living cells that serve as a source of new bone form in a process called osteogenesis [1].

Tissue can be engineered in several different ways, but there is a paradigm based on tissue regeneration (in vivo), which can serve as a template. In order to achieve this, it is crucial for the biomaterial to disintegrate as new tissue is formed. The ultimate goal of surgery from ancient times to the present day has always been the tissue repair of biological organs [4,5]. A variety of materials have been used over time to replace or generate bone tissue. Most large tissues and organs require support for the formation of new cells. The support is called a scaffold, template, or artificial extracellular matrix [6,7,8,9,10,11,12,13], and its main role is to regenerate bone tissue.Cells in porous matrices are expected to undergo an inflammatory phase, and thus proliferate and differentiate in the inflammatory phase [14,15].

In the second stage, biodegradable alloys are implanted into the appropriate anatomical site, and in vivo remodeling aims to restore the tissue or organ [16,17]. The key processes that occur during the in vitro and in vivo stages of tissue formation and maturation are:−Proliferation and biodegradation organization of scaffold extracellular matrix;−Remodeling and potential tissue growth;−Proliferation (sorting and differentiation of cells);−Proliferation and organization of extracellular matrix in order to achieve the purpose of tissue reconstruction [18].

First, all scaffolds must be made of highly biocompatible materials so that any negative immune response can be observed in the host tissue after local implantation [19]. Surface roughness is also needed to facilitate the fixation of the cells. Artificial scaffolding should fuse with the host tissue without scar formation. Mechanical strength and high rigidity are required to resist shrinkage forces and subsequently reshape remodeled tissues [20,21].

Furthermore, the side effects of scaffold degradation must be non-toxic. Specifically, the rate of resorption should coincide as much as possible with the rate of bone formation (in a few months and a few years). As the scaffold degrades and decomposes, it leaves behind the newly formed tissue that will take over the mechanical load. In addition, scaffolding should be easy to manufacture in as many shapes and sizes as possible [19,20,21,22].

However, the impossibility of creating the so-called “ideal scaffold” for bone grafting should be mentioned. Being the eighth most abundant in the lithosphere and the second most abundant in the hydrosphere, magnesium is a widely distributed element in the natural world [23,24]. Jin et al. studied magnesium-based metallic glasses for usage as medical devices. In these studies, magnesium alloys are usually analyzed in the form of rods or plates. Cylindrical scaffold structure has also been used in Mg alloys. It was designed to reduce the used mass of the alloy for implantation in a shape imitating a cortical bone. Alternatively, including some specific alloying elements, such as Ca or Y, biodegradable Mg-based alloys properties are closer for orthopedic use. In vitro and in vivo tests have also validated the advantages of Mg–Ca-based alloys with other chemical elements with high cell viability and good osteogenesis activity. However, efforts are still required to overcome the existing challenges before final successful clinical applications [25].

Wang et al. [26] presented in their study research about magnesium alloys compared with natural bone. Magnesium alloys present higher mechanical strength, with the Young Modulus very close to the biological bone. This aspect summarizes unique properties in reducing stress shielding during stress transfer at the implant–bone interface. A 0.2% Ca addition decreases about 1/3 of the degradation rate of as-cast Mg-4Zn and Mg-0.5Ca-2Y alloy [27].

Yang et al. studied a Mg scaffold with spherical pores and a cambered pore strut, noticing that it presented a quicker degradation rate, but the implant showed better resistance to the deterioration of interface, as compared to the scaffold with irregular pores and polygonal strut [28].

Of note is the essential role that Mg plays both in chlorophyll molecules and in any reaction that requires adenosine triphosphate (ATP), which is of much importance in the use of energy [29,30]. Being the fourth most common element in the human intracellular body, it is not surprising that Sarie et al. claim that Mg is involved in over 300 known enzymatic reactions [31], half of the Mg found in the adult human body being stored in the bone tissue [32].

Once absorbed, the magnesium is transported by the circular system and taken up by the tissues. Only up to 5% of the filtered Mg is excreted in the urine, provided that the Mg is within normal limits. It is proven that bones are used as a deposit of Mg [32].

## 2. Materials and Methods

### 2.1. Production Processes

The manufacture of magnesium-based alloys is a difficult process due to the phenomenon of burning elemental Mg, as well as the losses involved in normal manufacturing processes. It is therefore necessary to be processed in a controlled environment, using devices specifically designed for this purpose, separately in an inert argon atmosphere [33].

The elaboration of the alloys from the Mg-Ca-Y system was constructed from the start using high-purity components: Mg-98.5% and master-alloys: Mg-Y (70 wt.%–30 wt.%) and Mg-Ca (85 wt.%–15 wt.%) [27].

### 2.2. Synthesis, Morphology, and Structure Analysis of Mg-Ca-Y Alloys

The main alloy used to make the samples was from Hunan China Co., Ltd. [34]; the chemical composition is shown in Table 1. Using rectangular rods as raw materials, the obtained micro-ingots were cut into spherical samples by heating them in an induction furnace (Inductro SA, Bucharest, Romania) at a temperature of 680–690 °C for 30 min under inert protective atmosphere Ar5.0, as shown in Table 1, with a diameter of 20 mm and a thickness of 2 mm. The raw material loading of the test sample was approximately 23 g/ingot. The samples were ground with a grinding paper with a particle size of 340–2000 MPi, polished with an alumina suspension (1–6 μm), washed with alcohol, then sonicated in ethanol for 10 min. The test samples were etched with magnesium acetate solution (CH_3_COO)_2_ × 4H_2_O for microstructural analysis. Surface morphology was analyzed with an optical microscope (Leica DMI 5000 microscope, Wetzlar, Germany) and a scanning electron microscope (SEM FEI Quanta 200 3D, dual beam, equipped with an energy-dispersive X-ray spectrometer Unit-X flash Bruker, (Harvard, Cambridge, MA, USA) [27]. Basic characterization of magnesium alloys, such as XRD and SEM analysis, are presented in previous research [30], where addition of a yttrium element in the Mg-Ca-Y alloys refines the microstructure. The XRD results show the formation of Mg_24_Y_5_ with a cubic structure. Yttrium phases present a white spherical aspect in the metallic matrix, with an average size of 16–20 µm. Calcium also exhibits a eutectic phase—Mg_2_Ca—which is located at the magnesium grains boundary.

### 2.3. Biocompatibility of Mg-Ca-Y Alloys

Samples in the form of flat metallic parts (weight between 0.68 g and 0.98 g) were ultrasonically cleaned with acetone–ethanol and exposed to ultraviolet (UV) light for 30 min on each side for sterilization. They were then placed in the middle of cell culture suspension. The inserts (0.4 µm pore size) used in the 24-well plate allowed alloy samples to be matched to cells to assess cell viability. In addition, the examined Mg-0.5Ca-xY alloy samples were immersed in complete medium at 37 °C and 5% CO_2_ for 5 days to assess pH changes during co-incubation (days 1, 3, and 5) [27].

In previous research, Istrate et al. studied the biocompatibility of Mg-0.5Ca-xY alloys and observed that these alloys present a cytocompatibility behavior. The biocompatibility rate at three days for Mg-Ca-Y alloys (Y = 1.5, 2.0, and 3.0 wt.%) was above 70%. The biocompatibility level decreased after five days with values between 50% and 70%, especially for the alloys with 1.5 wt.%, 2.0 wt.%, and 3.0 wt.% Y. These changes can be specific to the following factors: change of pH value and ion release from alloys [27].

### 2.4. In Vivo Animal Study—Surgical Model and Study Protocol

The trial runs for this study were conducted following the Guide for the Care and Use of Laboratory Animals, according to the European Union legislation regarding the use of animals. The animal model selection and the number of trial subjects included in a group for an observation point were determined based on the studies published regarding the testing of Mg alloys that evaluate the biocompatibility, corrosion, and other similar factors [35,36,37,38,39,40,41]. A total of 20 Sprague Dawley male rats, approximately 30 weeks old, weighing between 250 and 350 g, were randomly assigned into 5 groups of 4 trial subjects each.

Trial subjects’ acclimation was ensured through constant temperature of 22 ± 0.8 °C and humidity of 60 ± 10% with a 12 h circadian cycle. Trial subjects were examined daily during the 7-day quarantine. The animals were purchased from the Cantacuzino National Institute of Research and Development of Microbiology and Immunology (București, Romania). The size of Mg-0.5 Ca-xY alloy pins used in this trial is 10–13 mm length and 1.5–2 mm diameter. The rats received 2% xylazine administered before surgery and 1–2% isoflurane mix inhalator support through surgery. The intervention was performed on the left lumbar area and on the left hind limb of each subject.

To begin, the rats were shaved on both hind limbs and torso and disinfected with betadine (Figure 1a). A 3–4 cm initial incision was performed over the left dorsolateral femur, as shown in Figure 1b. The Mg-0.5Ca-xY alloy pins were introduced next to the exposed femur through a lateral approach on the middle part of the femur (Figure 1c). The suture was performed in continuous points on the muscular layer and in separated points on the skin (Figure 1d). A second incision, approx. 3–4 cm, was performed in the left flank area, as shown in Figure 1e. The suture was performed in continuous points on the muscular layer and in separated points on the skin (Figure 1f).

Post-operatory, the rats were kept in cages where the displacement change, food and water consumption, signs of infection, wound healing, gas bags presence, and abnormal posture were monitored.

Post-operatory X-ray imagistic exams (Intermedical Basic 4006, Itallia and Examion CR—Smart, Agfa Healthcare, N.V., Mortsel, Belgium) and CT (General Electric Medical systems, Lightspeed 16, Chicago, IL, USA) were performed on rats at 1, 2, 3, 4, and 8 weeks to determine the Mg alloy pin position. Before exams, 2% xylazine was administered i.m. for anesthesia.

The subjects were euthanized at 1, 2, 4, and 8 weeks after surgery, and the left thigh and lumbar vertebrae were collected with magnesium alloy rods for histological examination. Femoral tissue samples were fixed in 10% buffered formalin for 24 h, desalted with 5% trichloroacetic acid solution for 4 days, and embedded in paraffin after stereotyping. Specimens containing dorsal implants were fixed in 10% buffered formalin for 24 h, shaped, and then embedded in paraffin. All samples were cut into 5 microns and stained using the Trichrome Masson method. A Leica DM750 photon microscope (Tokyo, Japan) and LAS version 4.2.0 from Leica Microsystems (Switzerland) Ltd. (Heerbrugg, Switzerland) were used to examine the samples. All animals in the groups undergoing surgical experiments were euthanized according to a well-established protocol.

SEM exam samples were fixed in 2% glutaraldehyde phosphate buffer, dehydrated in successive alcohol baths (alcohol 30%, 50%, 70%, 90%, absolute alcohol, and acetone), freely air dried, and covered with silver. The sample scanning was conducted using SEM FEI Quanta 2003D, dual beam, equipped with energy-dispersive X-ray spectroscopy analysis unit—Xflash (Bruker, Harvard, MA, USA).

## 3. Results and Discussions

### 3.1. Clinical Results

Clinical observation results conducted at 24 h, 1, 2, 4 and 8 weeks are presented in Table 2. Gas bag presence was observed as a result of Mg-0.5Ca-xY alloy pin degradation causing deformations in the implant area.

At 24 h and at 1-week post-operatory exams, for all five types of alloys used, femoral and lumbar deformations could be observed. At 2 weeks post-operatory exams, femoral area deformations could be observed for all five types of alloys used. Lumbar area deformations were only present in Y3, Y4, and Y5 rat batches. At 4 weeks post-operatory exams, only the Y4 and Y5 rat batches presented femoral area deformations. At 8 weeks post-operatory exams, neither region presented any deformations.

### 3.2. Imagistic Results

X-ray examinations performed after 2, 4, and 8 weeks (Figure 2) showed the presence of air pockets around the Mg-0.5Ca-xY needles for all five concentrations. The air pockets in the soft tissues of the waist were larger than the thighs. CT imaging (Figure 3) also observed a reduction in cavitation around the Mg-0.5Ca-xY needle at 1, 2, 4, and 8 weeks for all five concentrations used.

X-ray and CT exams performed at 4 and 8 weeks revealed gas bags present in both the femoral and lumbar areas, even though modifications of form or volume were not visible.

### 3.3. Histological Results

Histological exams performed at 1, 2, 4, and 8 weeks revealed good compatibility for all used implants, confirmed through the moderated inflammatory reaction, with rare presence of macrophages, fibroblasts, collagen fibers, young collagen tissue formation, and new formation of capillaries. The conjunctival sac also formed well around the implant material.

Figure 4 shows the histologically staged evolution of the material present at the implantation site at intervals of one, two, four, and eight weeks after surgery. Aspect at one week: a—implanted alloy, large vacuoles delimited by unformed connective tissue; b—areas of implanted material partially resorbed, delimited by a peripheral fibrous reaction; c—moderate inflammatory reaction, rare macrophages, fibroblasts, and collagen fibers. Aspects at two weeks: d—resorption and gas release of implant material, voids bounded by connective tissue; e—moderate inflammatory response around implant material; f—Mg-0.5Ca-xY alloy deposits, partially absorbed, surrounded by fibroblasts, collagen fibers (young connective tissue), and newly formed capillaries; aspects at week 4: g—significant resorption of implant material with gas release, void space bounded by connective tissue; h—implant collagen formation around the implanted material, partially resorbed, also with newly formed blood vessels; i—the presence of connective tissue around the implant; appearance at eight weeks: j—partially resorbed area of implant material with surrounding fibrous reaction; k—large vacuolar spaces in the tissue due to tissue degradation/absorption of the materials used; l—well-formed ligamentous pockets around the implanted material.

The specimens were extracted from rats and analyzed using the SEM FEI Quanta 200 3D electron beam microscope, dual beam, in order to observe the degradation of the specimens. The images are presented in Figure 5.

The beginning of biointegration was noted by cell proliferation (small cell aggregates) on the surface of the implant material. No inflammatory response or tissue incompatibility with implant material was observed.

As can be seen in Figure 6, on the surface of the implanted material, there is a very large mass of cell aggregates, represented by macrophages and fibroblasts, involved in the biodegradation and synthesis of new connective tissue.

In the first image in Figure 5 and Figure 6, there are multiple fragments of free material. In Figure 6, the degradation is more accentuated at the periphery of the material (centripetal). In Figure 7b, the fragmentation of the nucleus is observed directly proportional to the genesis of the connective tissue (collagen fibers). In Figure 8, the material is extremely porous/uneven, covered by an important fibro-cellular mass. The rough appearance is given by the biodegradation and resorption initiated by local cells. In the last image, the surface is fractured and extremely uneven/porous, with many cell aggregates indicating a very good integration of the implanted material.

Figure 9 on the left side shows the neogenesis of connective tissue in which small fragments of unabsorbed material are observed. The porous character of the surface of the implanted material is accentuated as its integration and resorption take place. In the image on the right, the porous surface is covered by numerous fibroblasts responsible for the production of collagen fibers, which gradually replace the resorbed material.

## 4. Conclusions

In vivo testing of magnesium-based alloys with different concentrations of yttrium indicates very good tissue compatibility and very good tissue resorption over 8 weeks, which ensures the time required for the consolidation of a bone fracture fixed with such an alloy. The histological examination revealed at the implantation site a well-organized connective tissue that replaced the resorbed implant material. This fibrous tissue may represent, in the context of a fracture, the solid support of a later mineralized bone matrix.

In addition, electron microscopic examination revealed a very good adhesion on the surface of the alloy of the cells involved in the healing process (fibroblasts, osteoblasts, osteoclasts, macrophages, endothelial cells, etc.). The complete withdrawal of gas deposits resulting from tissue degradation of the tested alloy is a promising result for its use in the field of orthopedics.

Therefore, all the results obtained from in vitro and in vivo testing of yttrium-enriched magnesium-based alloys indicate that these can successfully replace the classic, non-absorbable materials used today in the field of orthopedics.

## Figures and Tables

**Figure 1 materials-15-05958-f001:**
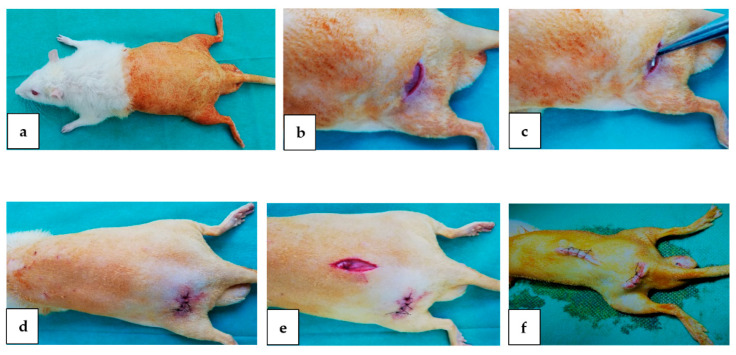
Surgical procedure used to implant metallic samples: (**a**)—preoperative aspect; (**b**)—the incision of the skin and the exposure of the femur; (**c**)—fixing the piece next the femur; (**d**)—skin suture; (**e**)—fixing the piece in the lumbar area; (**f**)—post-operatory appearance.

**Figure 2 materials-15-05958-f002:**
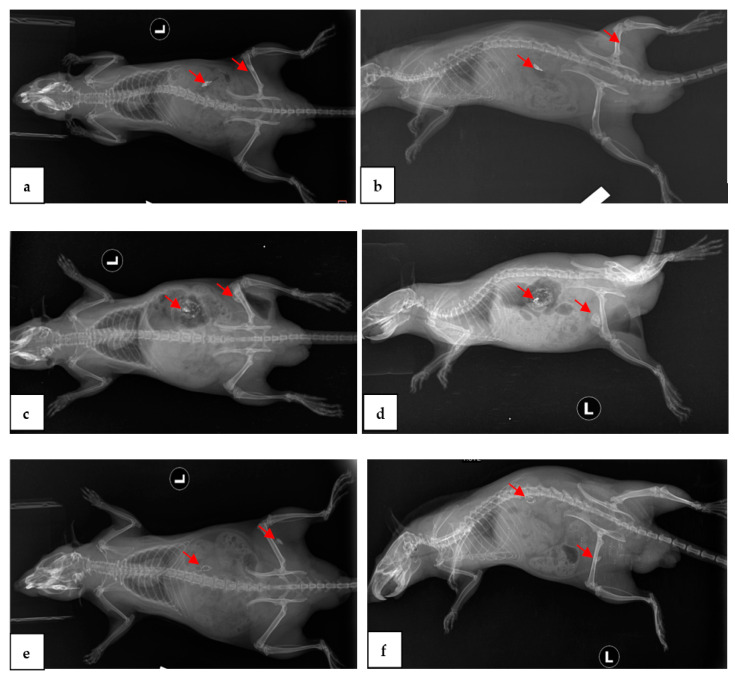
Rats, post-operative representative radiographs, Mg-0.5Ca-xY alloy (red arrows) and air sacs, dorsal–ventral (**a**,**c**,**e**) and lateral–lateral (**b**,**d**,**f**) lying. After 2 weeks (**a**,**b**), 4 weeks (**c**,**d**), and 8 weeks (**e**,**f**).

**Figure 3 materials-15-05958-f003:**
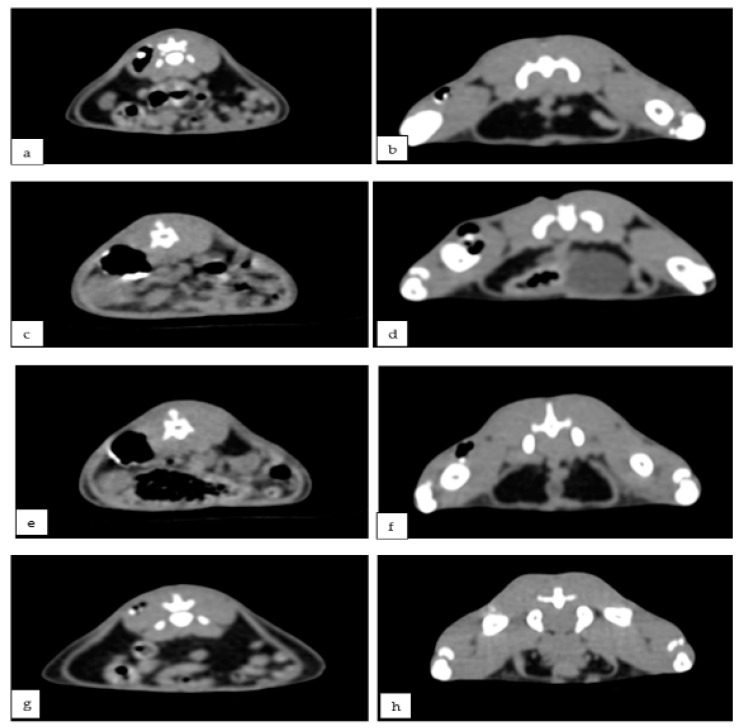
Rat, post-operative representative CT images, Mg-0.5Ca-xY alloy and gas inclusions, lumbar spine (**a**,**c**,**e**,**g**) and femur (**b**,**d**,**f**) regions. After 1 week (**a**,**b**), 2 weeks (**c**,**d**), 4 weeks (**e**,**f**), and 8 weeks (**g**,**h**).

**Figure 4 materials-15-05958-f004:**
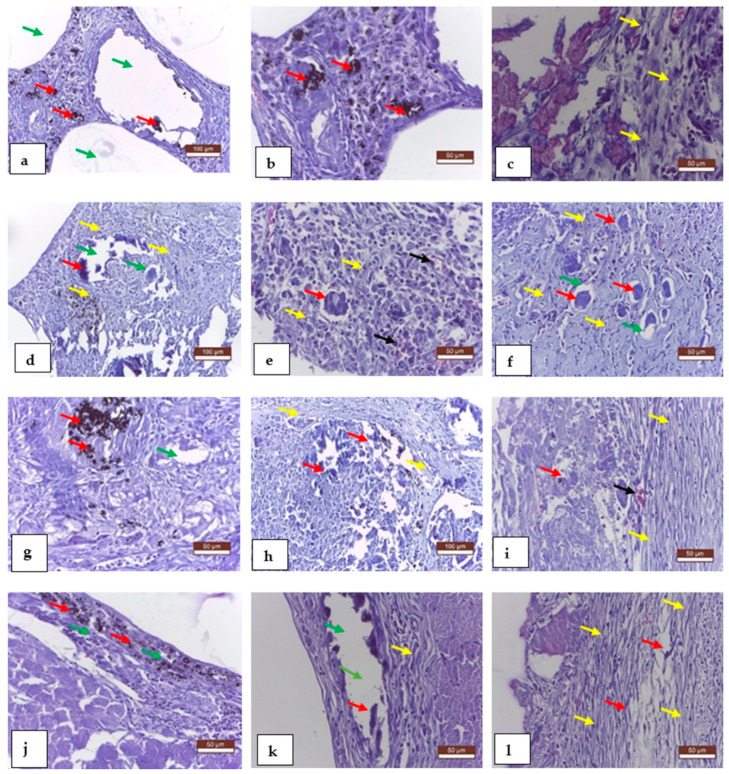
Rat tissue, representative results of histological examination, MTC staining, 1 week later (**a**–**c**), 2 weeks (**d**–**f**), 4 weeks (**g**–**i**), and 8 weeks (**j**–**l**). Mg-0.5Ca-xY alloy (red arrow), gas inclusions (green arrow), fibroblast collagen fibers (yellow arrow), and neovascularization (black arrow).

**Figure 5 materials-15-05958-f005:**
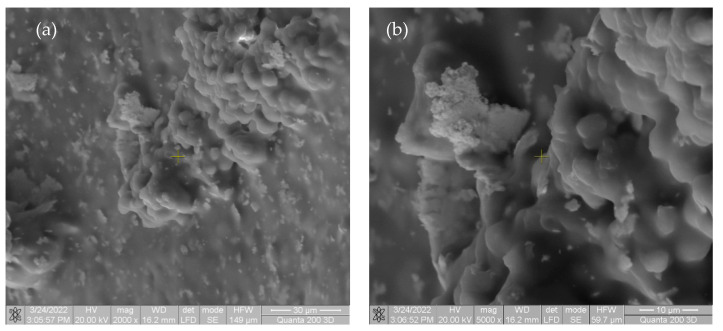
SEM images taken of rats after one week in (**a**) at magnification of 2000×; (**b**) at magnification of 5000×.

**Figure 6 materials-15-05958-f006:**
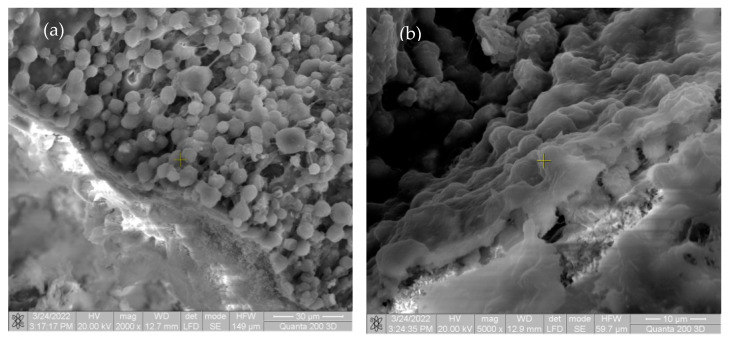
SEM images of rats after two weeks (**a**) at magnification of 2000×; (**b**) at magnification of 5000×.

**Figure 7 materials-15-05958-f007:**
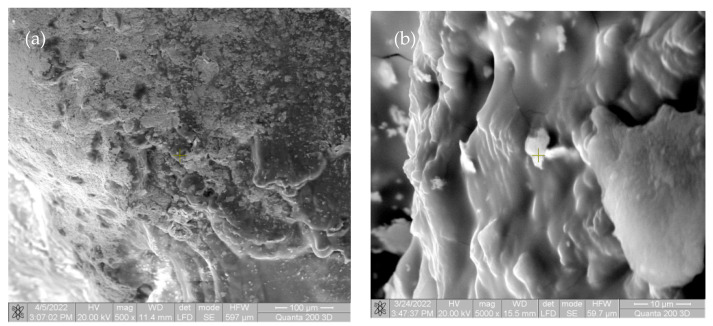
SEM images of rats after four weeks for Mg-0.5Ca-1.5Y (**a**) at magnification of 500×; (**b**) at magnification of 5000×.

**Figure 8 materials-15-05958-f008:**
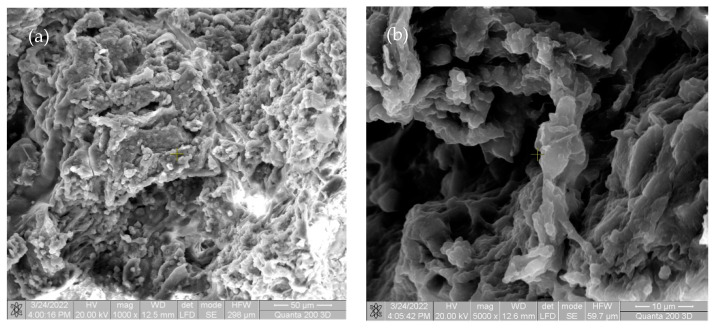
SEM images of rats after four weeks for Mg-0.5Ca-1.5Y (**a**) at magnification of 1000×; (**b**) at magnification of 5000×.

**Figure 9 materials-15-05958-f009:**
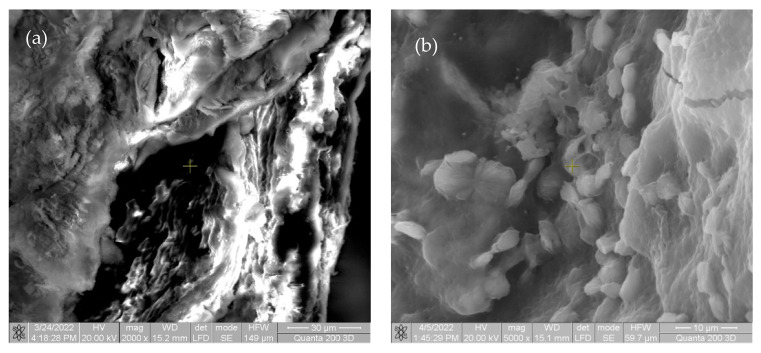
SEM images of rats after eight weeks (**a**) at magnification of 2000×; (**b**) at magnification of 5000×.

**Table 1 materials-15-05958-t001:** Load calculation for the 5 experimentally designed alloys [27].

*No.*	*Alloy*	*Mg [g]*	*Mg-15Ca [g]*	*Mg-30Y [g]*
1	Mg-0.5Ca-0.5Y	21.82	0.77	0.41
2	Mg-0.5Ca-1Y	21.42	0.77	0.82
3	Mg-0.5Ca-1.5Y	21	0.77	1.23
4	Mg-0.5Ca-2Y	20.59	0.77	1.64
5	Mg-0.5Ca-3Y	19.77	0.77	2.46

**Table 2 materials-15-05958-t002:** Local reaction, Mg-0.5Ca-xY alloy pins, clinical exam results.

	Lumbar Region	The Femoral Region
Implant	24 h	1 week	2 weeks	4 weeks	8 weeks	24 h	1 week	2 weeks	4 weeks	8 weeks
Y 2.1	+	+	-	-	-	+	+	+	-	-
Y 2.2	+	+	-	-	-	+ +	+	+	+	-
Y 2.3	+ +	+	+	+	-	+ +	+ +	+	-	-
Y 2.4	+	+	+	-	-	+ +	+	+	-	-
Y 2.5	+	+	-	-	-	+	-	-	-

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
