# Peer review of "Long-Term Examination of Degradation and In Vivo Biocompatibility of Some Mg-0.5Ca-xY Alloys in Sprague Dawley Rats"

_materials, 2022, doi:10.3390/ma15175958_

Round 1

Reviewer 1 Report

Stefan et al studied long-term examination of degradation and in vivo biocompatibility of Mg-0,5Ca-xY in Sprague Dawley. The results are interesting. However, the article is not well organized. The logic among paragraphs is confusing. The work cannot be accepted in its current form. The authors are advised to make the following changes.

1. Figures 1-3 are not seen in the manuscript. Basic characterization of magnesium alloys such as XRD and SEM should be provided.

2. The paragraph is fragmentary, the conditioning is not clear, the logical relation is confused, and the language needs to be reorganized.

3. What does the word "other" mean in the title? Strange choice of words. Please consider its rationality.

4. In the abstract and introduction part, it is necessary to clarify the key issues to be solved, the purpose of the paper, and the innovation.

5. The conclusion section should briefly introduce the main conclusions obtained, and the relevant background description should be condensed.

6. In recent years, the research papers on the degradability and bone implant of magnesium alloys need to be cited. The following papers are for the authors' reference.

[1] Biodegradable Mg–Zn–Ca-Based Metallic Glasses, Materials 2022, 15, 2172. https://doi.org/10.3390/ma15062172

[2] Biodegradable magnesium-based implants in orthopedics—A general review and perspectives, Adv. Sci. 2020, 7, 1902443.

[3] Mg bone implant: Features, developments and perspectives, Mater. Design. 2020, 185, 108259. 

Author Response

Dear Reviewer,

Thank you very much for your positive feedback. You ask to clarify more details on this manuscript, which certainly would give an improvement for our research. So, thank you for these valuable comments. We have tried to answer all aspects that were mentioned.

Stefan et al studied long-term examination of degradation and in vivo biocompatibility of Mg-0,5Ca-xY in Sprague Dawley. The results are interesting. However, the article is not well organized. The logic among paragraphs is confusing. The work cannot be accepted in its current form. The authors are advised to make the following changes.

  1. Figures 1-3 are not seen in the manuscript. Basic characterization of magnesium alloys such as XRD and SEM should be provided.

Dear Reviewer,

We had an error in manuscript preparation and we mismatched the number writing of the figures. Now the manuscript has the figures in order. Please accept the misunderstanding and the corrections made by us. Also, these materials have been studied in a Complex Project (national grant) and previous results have been published in other papers (See ref.: Istrate B., Munteanu C., Lupescu S., Chelariu R., Vlad MD., Vizureanu P., MATERIALS, Vol. 13, Is. 14, Article Number: 3082, 2020, DOI: 10.3390 / ma13143082, WOS: 000554140300001).

As summary:

Basic characterization of magnesium alloys such as XRD and SEM analysis are presented in previous research, where the addition of Yttrium element in the Mg-Ca-Y alloys refines the microstructure. The XRD results show the formation of Mg24Y5 with a cubic structure. Yttrium phases present a white spherical aspect in the metallic matrix, having an average size of 16-20 µm. Also, Calcium exhibits a eutectic phase—Mg2Ca, which is located at the Magnesium grains boundary.

  1. The paragraph is fragmentary, the conditioning is not clear, the logical relation is confused, and the language needs to be reorganized.

Dear Reviewer,

Thank you very much for your corrections. As you will going to see in the submitted revised paper, the authors have read and corrected the main parts of the manuscript inserting specific text in the results and discussions.

  1. What does the word "other" mean in the title? Strange choice of words. Please consider its rationality.

Dear Reviewer,

Thank you for your observations. We have modified the title” Long-term examination of degradation and in vivo biocompatibility of some Mg-0.5Ca-xY alloys in Sprague Dawley rats”

  1. In the abstract and introduction part, it is necessary to clarify the key issues to be solved, the purpose of the paper, and the innovation.

Dear Reviewer,

We appreciate the comments. The abstract was modified with the following text:

” The aim of the work is to test the biocompatibility of magnesium-based alloys on laboratory rats, in order to use these alloys in medical applications. The innovative part of this analysis is the chemical composition of the alloys used and the tests performed on laboratory animals.”

  1. The conclusion section should briefly introduce the main conclusions obtained, and the relevant background description should be condensed.

Dear Reviewer,

The conclusions part has been totally rephrased

„In vivo testing of magnesium-based alloys with different concentrations of Ytrium, indicates very good tissue compatibility and very good tissue resorption of over 8 weeks, which ensures the time required for the consolidation of a bone fracture fixed with such an alloy. The histological examination revealed at the implantation site a well-organized connective tissue that replaced the resorbed implant material. This fibrous tissue may represent, in the context of a fracture, the solid support of a later mineralized bone matrix. In addition, the electron microscopic examination reveals a very good adhesion on the surface of the alloy of the cells involved in the healing process (fibroblasts, osteoblasts, osteoclasts, macrophages, endothelial cells, etc.).

The complete withdrawal of gas deposits resulting from tissue degradation of the tested alloy is a promising result for its use in the field of orthopedics. Therefore, all the results obtained from in vitro and in vivo testing of ytrium-enriched magnesium-based alloys indicate that these can successfully replace the classic, non-absorbable materials used today in the field of orthopedics.”

  1. In recent years, the research papers on the degradability and bone implant of magnesium alloys need to be cited. The following papers are for the authors' reference.

[1] Biodegradable Mg–Zn–Ca-Based Metallic Glasses, Materials 2022, 15, 2172. https://doi.org/10.3390/ma15062172

[2] Biodegradable magnesium-based implants in orthopedics—A general review and perspectives, Adv. Sci. 2020, 7, 1902443.

[3] Mg bone implant: Features, developments and perspectives, Mater. Design. 2020, 185, 108259. 

Dear Reviewer,

Thank you very much for your observations. The mentioned papers are highly cited and their information have been added to the manuscript:

”Jin et. al studied magnesium-based metallic glasses for usage as medical devices. In main studies, magnesium alloys are usually analyzed in the form of a rod or plate. Also, a cylindrical scaffold structure has been used in Mg alloys. It was designed to reduce the used mass of the alloy for implantation in a shape imitating a cortical bone. As an alternative, including some specific alloying elements, like Ca or Y biodegradable Mg-based alloys are closer to orthopedic use. In vitro and in vivo tests have also validated the advantages of Mg–Ca-based alloys with other chemical elements, with high cell viability and good osteogenesis activity. However, efforts are still required to overcome the existing challenges, before final successful clinical applications (1).

Wang et. al presented in their study research about magnesium alloys compared with natural bone. Magnesium alloys present higher mechanical strength, with Young Modulus very close to the biological bone. This aspect summarizes unique properties in reducing stress shielding during stress transfer at the implant-bone interface. Also, 0.2% Ca addition decreases about 1/3 of the degradation rate of as-cast  Mg-4Zn  and Mg-0.5Ca-2Y alloy.(2)

Yang et. al studied an Mg scaffold with spherical pores and a cambered pore strut that presented a quicker degradation rate, but the implant showed better resistance to the deterioration of interface, as compared with the scaffold with irregular pores and polygonal strut. (3)”

The references have been inserted into the text.

Reviewer 2 Report

The manuscript "Long-term examination of degradation and in vivo biocompatibility of other Mg-0,5Ca-xY in Sprague Dawley" has been reviewed.

It is an experimental work on degradation and in-vivo biocompatibility of Mg-0.5Ca-xY alloy.

The manuscript can be reconsidered for publication at least after the following major revisions:

The novelty aspects of the research should be better highlighted in abstract and introduction.

Line 80: [19-22] not [1922]

Line 163: The, not “he”.

Figure 1-3 are not present in the manuscript and haven’t been discussed. The first figure is n. 4.

Figure 7 is not discussed in the manuscript.

Line 154: fig. 1 is called but is not present.

Figure 9 is not discussed in the manuscript.

Conclusions are weak and not completely supported by the results.

Author Response

Dear Reviewer,

Thank you very much for your positive feedback. You ask to clarify more details on this manuscript, which certainly would give an improvement for our research. So, thank you for these valuable comments. We have tried to answer all aspects that were mentioned.

The manuscript "Long-term examination of degradation and in vivo biocompatibility of other Mg-0,5Ca-xY in Sprague Dawley" has been reviewed.

It is an experimental work on the degradation and in-vivo biocompatibility of Mg-0.5Ca-xY alloy.

The manuscript can be reconsidered for publication at least after the following major revisions:

  • The novelty aspects of the research should be better highlighted in the abstract and introduction.

Dear Reviewer,

We appreciate the comments.

The abstract was modified with the following text:

” The aim of the work is to test the biocompatibility of magnesium-based alloys on laboratory rats, in order to use these alloys in medical applications. The innovative part of these analyses is the chemical composition of the alloys used and the tests performed on laboratory animals.”

The introduction was updated with the following text :

”Jin et. al studied magnesium-based metallic glasses for usage as medical devices. In main studies, magnesium alloys are usually analyzed in the form of a rod or plate. Also, a cylindrical scaffold structure has been used in Mg alloys. It was designed to reduce the used mass of the alloy for implantation in a shape imitating a cortical bone. As an alternative, including some specific alloying elements, like Ca or Y biodegradable Mg-based alloys are closer to orthopedic use. In vitro and in vivo tests have also validated the advantages of Mg–Ca-based alloys with other chemical elements, with high cell viability and good osteogenesis activity. However, efforts are still required to overcome the existing challenges, before final successful clinical applications (1).

Wang et. al presented in their study research about magnesium alloys compared with natural bone. Magnesium alloys present higher mechanical strength, with Young Modulus very close to the biological bone. This aspect summarizes unique properties in reducing stress shielding during stress transfer at the implant-bone interface. Also, 0.2%  Ca addition decreases about 1/3 of the degradation rate of as-cast  Mg-4Zn  and Mg-0.5Ca-2Y alloy.(2)

Yang et. al studied an Mg scaffold with spherical pores and a cambered pore strut that presented a quicker degradation rate, but the implant showed better resistance to the deterioration of interface, as compared with the scaffold with irregular pores and polygonal strut. (3)”

[1] Biodegradable Mg–Zn–Ca-Based Metallic Glasses, Materials 2022, 15, 2172. https://doi.org/10.3390/ma15062172

[2] Biodegradable magnesium-based implants in orthopedics—A general review and perspectives, Adv. Sci. 2020, 7, 1902443.

[3] Mg bone implant: Features, developments and perspectives, Mater. Design. 2020, 185, 108259.

  • Line 80: [19-22] not [1922]

Dear Reviewer,

We have corrected the reference format.

  • Line 163: The, not “he”.

Dear Reviewer,

We have corrected the mistake.

  • Figure 1-3 are not present in the manuscript and haven’t been discussed. The first figure is n. 4.

Dear Reviewer,

We had an error in manuscript preparation and we mismatched the number writing of the figures. Now the manuscript has the figures in order. Please accept the misunderstanding and the corrections made by us. Also, these materials have been studied in a Complex Project (national grant) and previous results have been published in other papers (See ref.: Istrate B., Munteanu C., Lupescu S., Chelariu R., Vlad MD., Vizureanu P., MATERIALS, Vol. 13, Is. 14, Article Number: 3082, 2020, DOI: 10.3390 / ma13143082, WOS: 000554140300001).

  • Figure 7 is not discussed in the manuscript.

Dear Reviewer,

Due to the error corrected above, the figure numbers are updated. Figure 7 now figures 4 and the explanation is the following:

Figure 4 shows the histological staged evolution of the material present at the implantation site at intervals of one, two, four, and eight weeks after surgery. Aspect at one week: a - Implanted alloy, large vacuoles delimited by unformed connective tissue; b - Areas of implanted material partially resorbed, delimited by a peripheral fibrous reaction; c - Moderate inflammatory reaction, rare macrophages, fibroblasts, and collagen fibers. Aspects at two weeks: d - resorption and gas release of implant material, voids bounded by connective tissue; e - moderate inflammatory response around im-plant material; f - Mg-0.5Ca-xY alloy deposits, partial Absorbed, surrounded by fibroblasts and collagen fibers (young connective tissue) and newly formed capillaries; aspects at week 4: g - significant resorption of implant material with the gas release, void space bounded by connective tissue; h - implant Collagen formation around the implanted material, partially resorbed, also with newly formed blood vessels; i - the presence of connective tissue around the implant; appearance at eight weeks: j - partially resorbed area of implant material with surrounding fibrous reaction; k - Large vacuolar spaces in the tissue due to tissue degradation/absorption of the materials used; l - Well-formed ligamentous pockets around the implanted material.

  • Line 154: fig. 1 is called but is not present.

Dear Reviewer,

We had an error in manuscript preparation and we mismatched the number writing of the figures. Now the manuscript has the figures in order. Please accept the misunderstanding and the corrections made by us.

All figures have now explanations.

  • Figure 9 is not discussed in the manuscript.

We had an error of manuscript preparation and we mismatched the number writing of the figures. Now the manuscript has the figures in order. Please accept the misunderstanding and the corrections made by us.

Figure 9 now figures 6 and below are the explanations:

The beginning of bio-integration, is noted by cell proliferation (small cell aggregates) on the surface of the implant material. No inflammatory response or tissue incompatibility with the implanted material was observed. As can be seen in figure 6, on the surface of the implanted material, there is a very large mass of cell aggregates, represented by macrophages, and fibroblasts, involved in the biodegradation and synthesis of new connective tissue.

  • Conclusions are weak and not completely supported by the results.

Dear Reviewer,

The conclusion part has been totally rephrased.

„In vivo testing of magnesium-based alloys with different concentrations of Yttrium, indicates very good tissue compatibility and very good tissue resorption of over 8 weeks, which ensures the time required for the consolidation of a bone fracture fixed with such an alloy. The histological examination revealed at the implantation site a well-organized connective tissue that replaced the resorbed implant material. This fibrous tissue may represent, in the context of a fracture, the solid support of a later mineralized bone matrix.

In addition, the electron microscopic examination reveals a very good adhesion on the surface of the alloy of the cells involved in the healing process (fibroblasts, osteoblasts, osteoclasts, macrophages, endothelial cells, etc.).

The complete withdrawal of gas deposits resulting from tissue degradation of the tested alloy is a promising result for its use in the field of orthopedics.

Therefore, all the results obtained from in vitro and in vivo testing of ytrium-enriched magnesium-based alloys indicate that these can successfully replace the classic, non-absorbable materials used today in the field of orthopedics.”

Round 2

Reviewer 1 Report

The authors addressed all of the reviewer's concerns and made changes as requested. In this case, I suggest that this article may be considered for acceptance.

Reviewer 2 Report

The manuscript has been significantly improved and can be accepted in the present form.